# An Enhanced Replica Selection Approach Based on Distance Constraint in ICN

Yaqin Song [1,2], Hong Ni [1,2] and Xiaoyong Zhu [1,2,*]

1 National Network New Media Engineering Research Center, Institute of Acoustics, Chinese Academy of Sciences No. 21, North Fourth Ring Road, Haidian District, Beijing 100190, China; songyq@dsp.ac.cn (Y.S.); nih@dsp.ac.cn (H.N.)
2 School of Electronic, Electrical and Communication Engineering, University of Chinese Academy of Sciences No. 19(A), Yuquan Road, Shijingshan District, Beijing 100049, China
* Correspondence: zhuxy@dsp.ac.cn; Tel.: +86-131-2116-8320

**Abstract:** Fifth generation (5G) networks have a high requirement for low latency of data delivery. Information-centric networking (ICN) adopts the paradigm of separation of the identifier and locator. It is efficient in content distribution by supporting in-network caching and has the potential to satisfy the low latency requirement in 5G. Replica selection is a key problem to retrieving content in ICN. Prior research usually utilizes the nearest replica. However, using the nearest replica cannot guarantee the smallest content download delay. To exploit in-network caching better, we propose an enhanced replica selection approach, called ERS. ERS first uses a distance-constrained-based name resolution system to discover the nearby replicas. Then, the most appropriate replica is chosen according to a local state table that maintains the state of replica nodes within a limited domain. In addition to network distance and replica node load, ERS innovatively introduces the path congestion degree between requester and replica nodes to assist replica selection. With extensive simulations, the proposed approach shows better performance than the state-of-the-art methods in terms of average content download delay. Finally, the overhead of the proposed method is analyzed.

**Keywords:** information-centric networking; replica selection; in-network caching; name resolution

## 1. Introduction

The original design of most Internet protocols focused on the host-to-host communication model for the exchange of information between two specific endpoints. However, usage patterns and supporting technologies of the Internet have greatly changed, and today's Internet usage is dominated by the distribution and retrieval of content instead of connecting to a specific server [1]. Additionally, applications are increasingly demanding high throughput and low latency, and users care more about the content itself than the source of this content. Therefore, inefficient content delivery increasingly occurs in the current Internet Protocol (IP) infrastructure. For example, a user often connects to a distant server to retrieve content even if there is a nearby copy. A significant amount of network traffic is also generated when many users download the same popular content repeatedly and independently [2]. These mismatches of traditional protocols and current usage patterns also result in several difficulties concerning mobility, multihoming, scalability, etc. [3].

Content Delivery Networks (CDN) and Peer-to-Peer networks (P2P) [4] are considered to be early attempts towards directing the focus of end-to-end communication on content retrieval. CDN and P2P provide retrieval mechanisms as overlays over the existing Internet infrastructure and exploit the availability of cheap storage and processing capabilities. Although CDN and P2P complement traditional network protocols, their performance is still limited due to their operation at the application layer [5]. The appearance of the fifth generation (5G) networks is expected to break the performance bottleneck of

current communication networks [6]. Applications of 5G networks also have critical requirements for the end-to-end latency, especially in the ultra-Reliable Low Latency Communication (uRLLC) scenario [7,8]. In order to realize the high-performance indicators in 5G, researchers are working hard on several supporting technologies, especially the optimization of network architecture.

A novel networking paradigm called information-centric networking (ICN) has been proposed [9,10]. It is under standardization and has the potential to enhance data delivering services in the 5G networks [11]. ICN changes the Internet communication paradigm from the host-centric model to the information-centric model. It utilizes a content identifier (ID, also called name) at the networking level to address and retrieve the corresponding content data. With in-network caching technology, ICN can provide significant benefits, such as faster content delivery, network traffic reduction, and reducing duplicated transmissions for the same content.

To take full advantage of in-network caching, a replica selection schema is needed to select the most appropriate replica, for which two problems need to be solved. One is how to discover the replica nodes holding the requested content with low traffic overhead. The other one is how to select the best replica among a list of replicas of this content, in order to reduce content download delay.

The typical ICN schema named NDN (named data networking) [3] relies on an opportunistic cache-hit (also called happen-to-meet) to utilize in-network caching. In this fashion, only replicas cached on the path towards the content source can be utilized, which limits the potential advantages of network-wide in-network caching. To discover off-path cached replicas, some approaches based on NDN architecture are proposed to explore the network by sending probes or using content announcement mechanisms [12–16]. In the MobilityFirst schema (MF) [17], a global name resolution system (GNRS) is proposed to maintain mappings between IDs and network addresses (NA) of all cached contents, but scalability is the main problem [18]. After obtaining the NA list of replica nodes holding requested content from the name resolution system (NRS), many approaches choose the nearest replica node to retrieve this content [19]. However, the nearest replica is not guaranteed to be the one to which the user has the highest transmission throughput and the smallest content download delay [20].

In this paper, we propose a novel replica selection approach in ICN, denoted as ERS (enhanced replica selection). To resolve the content discovery problem, ERS is designed to leverage a distance-constrained-based enhanced name resolution system (ENRS) to discover the nearby replicas and a GNRS [17] to guarantee reachability. Additionally, a replica node state table (NST) is maintained locally to select the best replica among replica nodes within the domain limited by ENRS. By choosing replicas according to NST, ERS is able to reduce average content download delay, promote load balancing between replica nodes, and avoid data transmission along the congestion paths.

The main contributions of our work can be summarized as follows:

(1) We propose an enhanced replica selection approach called ERS to realize efficient content retrieval in the ICN scenario. ERS leverages a distance-constrained-based name resolution system to discover all nearby cached replicas, and the most appropriate replica is chosen according to a local node state table that maintains the state of replica nodes within a limited domain.

(2) We innovatively introduce path congestion degree as one of the main factors to select an appropriate replica. The detailed method to maintain the information of these factors and update the node state table is designed. Meanwhile, a refresh mechanism to avoid information expire is also adopted.

(3) We conduct sufficient simulation experiments to measure the performance of our approach. We compare our approach with several state-of-the-art methods, and the results show that our approach has better performance in terms of average content download delay and performs effectively in server and link load balance. Finally,

we analyze the overhead of ERS and further prove the availability of the proposed approach.

The remainder of this paper is organized as follows. We survey related work about content discovery mechanisms and replica selection strategies in Section 2. The design of ERS and the usage of NST are described, respectively, in Sections 3 and 4. In Section 5, we present the performance evaluation results and discuss the overhead of our proposal. Finally, we conclude our work in Section 6.

## 2. Related Work

In ICN, content is dynamically cached in routers, and replicas of the same content may be stored in multiple router nodes; thus, user requests can be responded to by a nearby cached replica instead of the distant content source, which reduces the transmission delay as well as saves the network bandwidth. To make full use of in-network caching, there have been several attempts at content discovery and replica selection.

### 2.1. Content Discovery Mechanisms

Identifier and locator split is among the most important characteristics in ICN architecture. Content is identified with a location-independent name that cannot be directly used for forwarding or routing purposes with the current IP network. As a result, an infrastructure that stores mappings between ID and NA is needed, which is named as name resolution service. In general, there are two major kinds of name resolution service approaches, the name-based routing approach and standalone name resolution approach [21].

In the name-based routing approach, the name resolution process is coupled with message routing. A typical example of this approach is NDN [3]. However, NDN only utilizes the cached contents along the routing path, which limits the network-wide usage of in-network caching. Thus, several works focus on utilizing off-path cached contents. DENA [12] provides shortcut paths for interest packets to access targeted content data packets under instructions of the announcement table constructed by a precision cached content announcement algorithm. In NRR [13], the first content request packet is used to explore the network, and the exploration is based on scoped flooding, limited by a radius using a TTL field in the packet. Lee et al. [2] proposed the SCAN scheme, in which, when the interest packet is forwarded to the content source, the intermediate router node sends the probe packet to explore the nearby cached replicas. However, SCAN needs convergence time to obtain a global view of the cached contents in a part of the network, and interest packets are always sent towards the content source and routed to nearby copies only if the request that just arrived was previously sent using the original Forwarding Information Base DIVER [14] compresses contents related to the name prefix extracted from the probe request and only sends an interest packet towards the producer if no Bloom Filter is present or the membership query fails. Under O2CEMF [15], each network node reactively explores nearby off-path caches when users begin to request a generation of content and then maintains the reachability information at the more coarse-grained generation level of granularity instead. With INFORM [16], the forwarding algorithm adaptively discovers volatile cached contents using content download delay feedbacks and a Q-learning technique.

In the standalone name resolution approach, the name resolution process and message routing process are decoupled. Such ICN approaches are proposed in MF [17], DONA [22], PSIRP [23], and NetInf [24]. For example, MF uses GNRS which provides the dynamic binding between a content name and its current replicas' NAes (network addresses), and the delivery of content can be realized only when the name resolution process is completed. However, maintaining the name mappings of all contents in the network is not feasible in terms of scalability, since the number of contents in the network is enormous [2].

Considering that applications have different requirements for the end-to-end latency, a multiple-level NRS was proposed in [25] that provides the relationship between a name

and replicas' NAes with the constraints of distances to achieve deterministic low latencies in a limited domain to accelerate the name resolution process.

### 2.2. Replica Selection Strategies

To select the best replica from a list of cached replica nodes, widespread research has been conducted, and several strategies have been proposed in both ICN content delivery and distributed key-value store application scenarios.

Prior research on ICN usually utilizes the nearest replica. For instance, MF uses GNRS to obtain replicas' locators, and then the access router looks up the closest cache location from the routing table and forwards the query to the applicable network [17]. However, the nearest replica is not guaranteed to be the one for which the requester has the smallest download delay [20]. In [26], the authors mentioned that NetInf first locates all available replicas and then selects the best one based on the network distance, delay, or other factors. However, the authors did not propose any detailed replica selection methods.

In current large-scale distributed key-value store systems for cloud computing, the value of each key is typically replicated and distributed across a group of replica servers. The client can select any one of these replica servers for each key-value access. The replica selection algorithms for the key-value store systems can be classified into three categories: information-agnostic, client-independence, and feedback according to their demanded information [27].

The information-agnostic algorithms do not need any extra information and are easy to implement. The representative methods are fixed, random, and round-robin strategies. However, a static replica selection strategy is not optimal [28]. The client-independence algorithms have the characteristic of utilizing information measured by the client independently. It uses some selection criteria, such as round trip time (RTT), the response time (RPT), outstanding requests (OSR), or bandwidth, to directly select one or more replica servers with relatively good performance. For instance, MongoDB selects the nearest replica servers by RTT at first, and then randomly chooses one of them [27]. The L2 [27] algorithm first chooses a subset of the replica servers with the lowest number of OSR, then the least RPT algorithm comes into force. The feedback algorithms have the characteristic of piggybacking information with the response from the server in each key-value access. A typical example of this category is C3 [29]. C3 determines the fastest replica server by its service time and the queue-size of the waiting requests. Furthermore, there are several methods using content transfer history to choose replicas directly to shorten content download delay by skipping the step of content search. The two-phase replica selection algorithm [30] uses a local file that records the historical content request of each user and then selects the best site with the highest score according to the user's quality of service and the context of the network. Rahman [31] predicts the minimal transport time site through a k-nearest neighbor algorithm according to file transfer history. However, this method only works effectively when the content request is contiguous. In [32], a replica selection algorithm based on node service capability for the edge cloud environment was proposed. Firstly, the service capability of the node is calculated based on the number of user requests and content download time in the replica node. Secondly, the service capability of all nodes containing the replica of the user request is evaluated, and the node with the best capability is returned to the user by the central cloud. However, this method is based on a central cloud, and the status evaluation is too simple to accurately assess node performance.

Inspired by the above research, we provided a replica selection approach, called ERS. ERS first obtains an NA list of replica nodes through ENRS and then uses a local node state table to select a replica with the best state. The node state is obtained by loads of replica nodes, network distance, and path congestion degree between the requester and the replica node. Finally, as shown in Table 1, we present a summary of the replica selection approaches in the literature.

**Table 1.** Summary of the replica selection approaches in the literature.

| Replica Selection Approach | Scenario | Content Discovery Mechanism | Replica Selection Strategy |
|---|---|---|---|
| NDN [3] | ICN (Information-Centric Networking) | Name-based routing, without name resolution | Utilize the nearest on-path replicas towards the content source |
| DENA [12] | ICN | Name-based routing, without name resolution | Using announcement table to utilize the nearest off-path replicas |
| SCAN [2] | ICN | Name-based routing, without name resolution | Intermediate routers send probe packets to explore the nearest off-path replicas |
| O2CEMF [15] | ICN | Name-based routing, without name resolution | Each network node explores nearby off-path replicas when users begin to request data |
| MF (MobilityFirst) [17] | ICN | Standalone name resolution, global name resolution | Global nearest replicas without considering link congestion or replica node overload |
| Random [27] | key-value stores | – | Randomly select replica, good balance among replica nodes, but may select a farther one |
| L2 [28] | key-value stores | – | First select replicas with the least number of outstanding requests, then select by the least response time |
| ERS (enhanced replica selection) proposed in this paper | ICN | Hybrid name resolution ofenhanced name resolution system and global name resolution system | Replicas a selected based on the distance constraint and replica node state |

## 3. ERS Design

In this section, we first give an example and model a content transmission process to clarify the motivation of our proposed ERS approach. Then, we provide an overview of ERS and introduce the pivotal components of ERS, including the content discovery mechanism based on distance constraint and the replica selection strategy based on a node state table.

### 3.1. Motivation

ICN supports the separation of identifier and locator. The user sends the request packet with a content name into the network, and the intermediate router which holds the corresponding content data immediately responds to the user via response packet. Consider a simple topology of a hybrid network of IP routers and cache-supported ICN routers, as shown in Figure 1 [33]. Supposing that a user requests content with its name, and the replica nodes holding this requested content include the content source and ICN routers (NA1, NA2, and NA3), then there are two main methods to retrieve this content. One is a happen-to-meet fashion like NDN, and the user will obtain the content data from the first replica node NA2, since it is on the transmission path to the content source. The other method is to use a GNRS to find all replicas and choose the nearest replica node as MF does, and the nearest replica node NA2 will be chosen as well. However, request packets from users connected to NA5 will always be forwarded to NA2 to retrieve content data, and replicas on NA3 will never be used. This may lead to high content download delay when the replica server at NA2 is in an overload situation or the available bandwidth between the link of NA2 and NA5 is very low. Therefore, in addition to considering the network distance, it is also important to consider load balancing between replica nodes and avoid data transmission on congested paths. Unfortunately, the optimal replica selection

strategy from a global optimization view is an NP-hard problem [34] and is hard to realize in content retrieval scenarios in ICN.

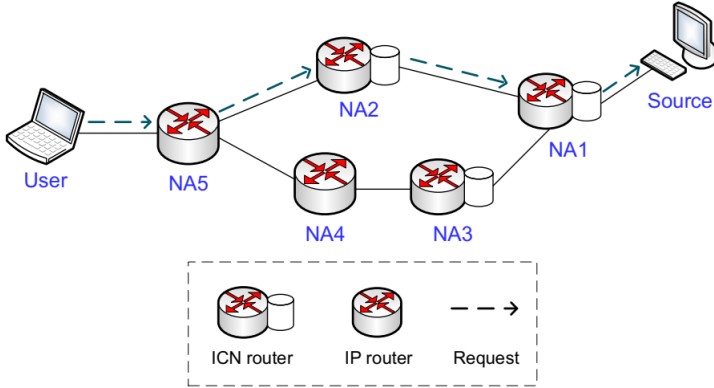

**Figure 1.** An example of content retrieval in ICN.

To analyze the impact factors of content download delay in ICN, we model the content transmission process, and the main symbols are listed in Table 2. Content download delay represents the time from the user sending a content request out to receiving the entire content data back. All the components of end-to-end delay include propagation delay, transmission delay, processing delay, and queuing delay [35]. Because the size of the content request packet is much smaller than that of the content response packet, we ignore the processing, transmission, and queuing delay of the content request packet. When a packet is put into the sending queue of one intermediate router, the time it stays in the queue is computed as the ratio of queue size over the bandwidth of the sending link at that time [36]. The queue size can be estimated by multiplying the number of flows by the size of each content. Thus, the queuing delay of a router can be measured as $n_r^q \times C / B_r^q$. As for the transmission delay in a replica node, it can be obtained by the size of the content divided by the available bandwidth. All flows share the network bandwidth [32]; thus, the available bandwidth is $B_s / n_s$. Therefore, the transmission delay of the replica node is $n_s \times C / B_s$. Additionally, the propagation delay is calculated by the accumulation of delay in all path links. Supposing that the CPU (central processing unit), memory, and other physical configuration is not bottlenecked, the processing delay can be neglected.

$$
\begin{aligned}
D_{end} &= D_{transmission} + D_{propagation} + D_{processing} + D_{queuing} \\
&= n_s \times \frac{C}{B_s} + 2 \times \sum_l d_{prop}^l + \sum_q n_r^q \times \frac{C}{B_r^q}
\end{aligned}
\tag{1}
$$

where $D_{end}$ is the end-to-end latency. From Equation (1), we can determine that the content download delay is not only relative to the propagation delay caused by the network distance but also relative to the queuing delay brought by the queue size of the router and the delay caused by loads of the replica node. This analysis inspired us to consider all of these factors when selecting replicas, especially the delay associated with link and replica node load.

**Table 2.** The meaning of symbols.

| Symbol | Meaning | Symbol | Meaning |
|---|---|---|---|
| $C$ | the size of a content | $B_s$ | the bandwidth of a sender |
| $d_{prop}^l$ | the link delay of link $l$ between two network nodes | $n_s$ | the number of flows in a selected replica node when a request arrives |
| $B_r^q$ | the bandwidth of router $q$ | $n_r^q$ | the number of flows in router $q$ |

### 3.2. Overview of ERS

The ERS approach is mainly applied to the ICN architecture that uses the standalone name resolution approach. The user sends a name resolution request to the NRS to obtain a resolution response containing an NA list of replica nodes; then, the user locally selects a suitable one based on its historical knowledge and triggers the download of the content, and the subsequent transport is running by NA routing. Compared with the other strategies to select the appropriate NA from the list of NAes for a certain name mentioned in [37], this method is compatible with IP infrastructure and avoids excessive processing burden on the NRS. In addition, ERS also makes two enhancements based on the above name resolution and replica selection approach.

Among the enhancements is the usage of ENRS. It is a general phenomenon in the network that the selected replica is more likely to appear around the user, and the remote replica node is less likely to be selected. Hence, we only maintain the state of replica nodes within a limited domain to maximize benefits while reducing the information maintenance costs. The division of this domain is due to the distance constraint proposed in [25], and it does not introduce too much extra overhead for replica selection. The other enhancement is that ICN routers can execute the operation of the name resolution query and replica selection. This is because end-users outside the region of the ICN network cannot obtain NAes of the in-network replica nodes for security. Thus, the edge router [38,39] is connected to aggregate user requests and locally maintains an NST with regard to replica nodes and corresponding state values. It directly chooses a node with the best state once it obtains a list of replica nodes from ENRS.

Figure 2 shows an overview of the ERS approach. The dotted oval represents the range of an ENRS domain based on distance constraints, and GNRS is usually deployed in the cloud and not drawn in this figure. Our replica selection method is supported by the late-binding technique [25] that any intermediate network nodes could initiate a name resolution request to the NRS and change the destination address of a packet in the hop-by-hop forwarding processes of this packet. A complete work process example of ERS is as follows: First, user1 sends a content request to an edge router at NA5 (step 1); NA5 sends a name resolution request with the content ID to ENRS1 to obtain the NA of the content replicas, but the query result is empty (step 2); thus, it sends a request packet to the content source, which is resolved by GNRS (step 3) and user1 can receive the response content. Meanwhile, this content is cached in NA1, NA3, and NA5 along the transmission path with the Leave Cache Everywhere cache algorithm [2,40], and the content name and these NAes are registered into corresponding NRS (step 4). Later, the edge router at NA4 receives a request for the same content from user2 (step 5); then, NA4 obtains NA3 and NA5 through name resolution by ENRS1 (step 6). Finally, the NST on the edge router at NA4 shows that NA3 has a better state with a smaller value compared to NA5, and the content request is responded to by the content replica on NA3 (step 7).

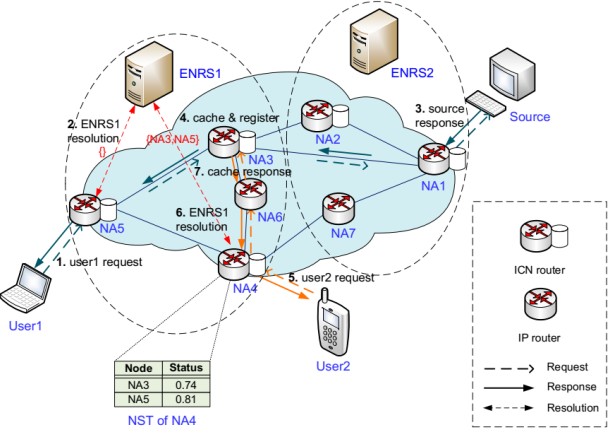

**Figure 2.** Overview of ERS approach.

The detailed operation process of replica selection performed by the edge router is presented in Algorithm 1. $N = \{NA_1, NA_2, NA_3, \cdots, NA_n\}$ is the addressset of replica nodes, where $n$ represents the number of replica nodes in the network. For the sake of intuition, the replica node is represented by its NA. $R$ is the subset of replica nodes resolved from ENRS. Then, the replica node $NA_{chosen}$ with the best state (i.e., smallest value) is returned to retrieve the requested content. It is worth noting that if the query result $R$ from ENRS is empty, then the query operation from GNRS will be performed, and the nearest replica selection strategy comes into force. It is an option that the ENRS and GNRS queries can be parallel, and the user can make a replica selection in the name resolution result of the faster response. If both the name resolution results from ENRS and GNRS are empty, it means that the requested content cannot be discovered in this ICN network. Additionally, the selected replica node could not effectively provide the data service because cached content on the replica node could be replaced or network congestion could occur in the network. Then, ICN routers can use the late-binding technique to reselect a replica node to complete this content delivery.

---

**Algorithm 1: The Operation Process of ERS**

---

**Input:** *contentID, NST*
**Output:** $NA_{chosen}$
1:   **initialization:** $R \leftarrow \varnothing$, $NA_{chosen} \leftarrow null$
2:   $R \leftarrow getNameResolutionByNRS(ENRS, contentID)$
3:   **if** $R \neq \varnothing$ **then**
4:       $NA_{chosen} \leftarrow chooseBestStateNA(R, NST)$
5:   **else**
6:       $R \leftarrow getNameResolutionByNRS(GNRS, contentID)$
7:       **if** $R \neq \varnothing$ **then**
8:          $NA_{chosen} \leftarrow chooseNearestNA(R)$
9:       **end if**
10:  **end if**
11:  **return** $NA_{chosen}$

---

### 3.3. Content Discovery Mechanism Based on Distance Constraint

Among the key features of ERS is leveraging both an ENRS based on distance constraint to improve efficiency and a GNRS to guarantee reachability. The goal of leveraging ENRS is to find the best replica node, not only the nearby copy to reduce the download delay but also to keep a low information maintenance overhead. If the ENRS cannot obtain the requested data, the content retrieval is guaranteed by GNRS that manages all copies as a fallback.

In [25], the authors mentioned a hierarchical ENRS framework with 3 layers, and it classified the end-to-end latency of 5G typical scenarios into three orders of magnitude: 0~1 ms, 1~10 ms, 10~100 ms [7,8]. In this multilayered ENRS scenario, it is recommended to adopt the random replica strategy, as mentioned in [41], when there are a small number of replica nodes at the lowest level. The number of replica nodes is moderate at the remaining levels, which is the main case to study in this paper, and is adopted in our proposed ERS strategy. A further discussion on the question about the delay range of the ENRS domain is in Section 5.3.

In this paper, a single-layer ENRS scenario is enough for us to study and validate our replica selection approach. As shown in Figure 3, the ENRS quantifiably organizes the network nodes into different domains by deterministic latency constraints. Every node is partitioned into an ENRS domain, and there is no area overlap between domains. Moreover, the GNRS, which keeps the global information of names, is responsible for guaranteeing that any name registered in GNRS is accessible and addressable. For a given resolution request of a certain content name, the ENRS resolution node returns an NA list of replica nodes holding the content within the ENRS domain.

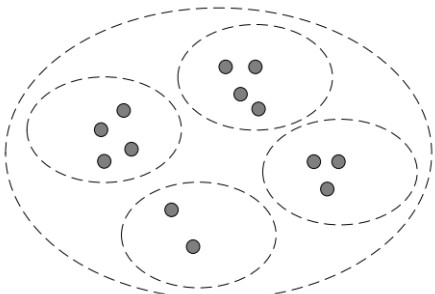

**Figure 3.** The structure of single-level ENRS. The grey points represent network nodes. The small dotted oval represents domains of ENRS, and the big dotted oval represents the domain of GNRS.

*3.4. Replica Selection Strategy Based on Node State Table*

After obtaining the name resolution result from NRS, ERS uses NST to select the most appropriate replica node from multiple replica nodes holding the requested content. NST maintains the state of the replica nodes within the ENRS domain. The simple structure of the NST enables quick replica selection decisions with little effect on content download delay. Figure 4a shows that NST is maintained on the edge router locally. Because all the content requests from users and all the content responses from replicas have to forward across the edge router, the edge router has the ability to record each content transmission information in a local log file (denoted as LOG). This information for different content transmission processes can be easily distinguished by the name in the networking layer.

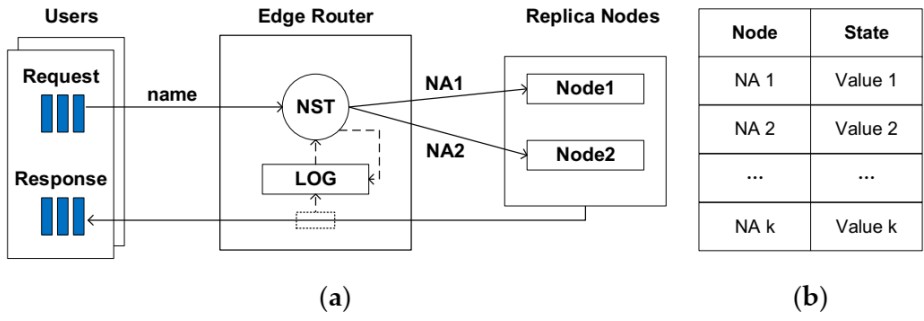

(a)                                (b)

**Figure 4.** The mechanism of replica selection based on node state table (NST) includes (**a**) the working schema of NST and (**b**) the data structure of NST.

The data structure of NST is shown in Figure 4b. The match field uses NA to distinguish replica nodes, and the state value indicates the ability to provide content service by the corresponding replica node. The calculation of state value is defined as Equation (2):

$$\text{value} = F(X), \ X = \ (x_1, x_2, \cdots, x_h) \tag{2}$$

where X is the network information collected during each content delivery by the edge router, such as the network distance, loads of replica nodes, and congestion degree of transmission paths. F(X) can be flexibly defined based on the optimization goal.

## 4. Usage of Node State Table

To select the best replica, each edge router maintains the information of replica nodes within its ENRS domain. Since the number of replica nodes in the network can be huge and the node state is dynamically changing, the usage and maintenance of NST is important and has a direct impact on the performance of content retrieval. In this section, we describe in detail how to calculate the node state values in NST and how to update the information used by NST.

### 4.1. Calculation of Node State Values

To better evaluate the state of replica nodes to achieve better network performance, not only classical information, such as distance and load balancing, among replica nodes are considered. We also innovatively introduce the concept of congestion degree for replica selection to mitigate network congestion caused by link overload.

Firstly, the performance of the replica selection is highly dependent on the distance between the replica node and the requester (i.e., edge router). The network distance (i.e., Hops) is evaluated by a traditional topology-based routing scheme based on shortest path algorithms [42]. If the network is built based on an IP-compatible infrastructure, the network distance can also be obtained by initializing the TTL field of IPv4 or Hop Limit field of IPv6 field to 255 on the sender side, and calculating the difference value between the value of the field received and 255 on the receiver side. This method is adopted in ERS because the operation is supported in both the IP Router and ICN Router, and no extra overhead is involved.

Secondly, we use a distributed method to evaluate the load of replica nodes, which is calculated using the outstanding requests (OSR) indicator. The OSR represents the outstanding requests sent by the edge router to the corresponding replica node, and this can reflect the load degree of a replica node to a great extent. By acquiring node load in a distributed way, the maintenance of load information can be accomplished with as little overhead as possible to avoid the introduction of a new central structure or extra burden on replica nodes for supporting the calculation of loads.

Thirdly, the congestion degree of the path between the edge router and the replica node is considered to avoid data transmission on a congestion path. When the total size of data packets is higher than the capacity of a router queue, it causes congestion, which results in dropping packets or large queuing delay and leads to an unacceptable long end-to-end latency [43]. Thus, it is necessary to make an advanced prediction of congestion [44] and avoid it. In ERS, we introduce the congestion degree metric for saving end-to-end latency. The inspiration for the method to evaluate congestion degree comes from the DCP algorithm proposed in [45]. Neglecting the processing delay, propagation, and transmission delays since they are constant components, the only variable component is the queuing delay caused by link congestion [35]. The DCP estimates queuing delay as follows:

$$delay_{queuing} = delay_{current} - delay_{base} \tag{3}$$

where the $delay_{base}$ is the minimum delay that packets experience along the transmission path. The packets delay measured in DCP uses the one-way delay as Equation (4):

$$delay_p = A_p - S_p \tag{4}$$

where $A$ is the arrival time for a packet and $S$ is the starting time of that packet. Although there is a time synchronization problem in the calculation of the one-way delay, the measurement errors in delays due to clock offset are canceled out by their difference in the queuing delay estimate. For example, the one-way delay of packet 1 is $delay_{p1} = A_{p1} - S_{p1}$, and that of packet 2 is $delay_{p2} = A_{p2} - S_{p2}$; supposing the one-way delay of packet 1 is smaller than that of packet 2, the queuing delay equals $(A_{p2} - S_{p2}) - (A_{p1} - S_{p1})$ according to Equation (3). Then, the queuing delay can be calculated as $(A_{p2} - A_{p1}) - (S_{p2} - S_{p1})$, which has no synchronization problem. However, the calculation of this one-way delay needs the packet sender, i.e., the replica node, to stamp the timestamp of this packet in its packet header. The conception of queueing delay is also used as an implementation of congestion pricing in [37], and the queuing delay is obtained by accumulating the queueing delay of every router, which brings extra overhead to the intermediate routers. Different from the accumulating method, queuing delay in ERS is measured as $RTT - RTT_{base}$ like

with TCP Vegas [46] and FAST [47], as shown in Equation (5), where *RTT* is the current RTT, and $RTT_{base}$ is the shortest RTT over a period of time.

$$delay_{queuing} = RTT - RTT_{base} \tag{5}$$

Finally, the node state value is calculated using Equation (6), where $w_i$ (i = 1, 2, 3) represents the weight assigned to each attribute. The network distance, loads of replica nodes, and path congestion degree are denoted by $x_1$, $x_2$, and $x_3$, respectively. Since these attributes have different physical characteristics, we need to normalize them into [0, 1] using the unity-based normalization [48]. Equation (7) is used to implement a unity-based normalization, also called max–min standardization, where $X_i$ represents each sampled data value; $X_{min}$ *and* $X_{max}$ are the minimum and maximum value among all the data values, respectively.

$$\text{Value} = \sum_{i=1}^{3} w_i \times x_i, \ w_i > 0, \ \sum_{i=1}^{3} w_i = 1 \tag{6}$$

$$X_i = \frac{X_i - X_{min}}{X_{max} - X_{min}} \tag{7}$$

To tackle users' various QoS and determine the $w_i$ (i = 1, 2, 3), the analytic hierarchy process method [30] is introduced. Build a comparison matrix C as in Table 3, where the $a_{ij}$ (j = 1,2,3) is chosen from a 1–9 ratio scale, as in Table 4, and $a_{ii}$ equals one. Thus, the vector $\overrightarrow{w}$ can be calculated using C.

**Table 3.** Comparison matrix C.

| C | $x_1$ | $x_2$ | $x_3$ |
|---|---|---|---|
| $x_1$ | $a_{11}$ | $a_{12}$ | $a_{13}$ |
| $x_2$ | $a_{21}$ | $a_{22}$ | $a_{23}$ |
| $x_3$ | $a_{31}$ | $a_{32}$ | $a_{33}$ |

**Table 4.** 1–9 ratio scale.

| Importance | Definition | $a_{ij}$ |
|---|---|---|
| Equal | $x_i$ is important as $x_j$ | 1 |
| Moderate | $x_i$ is a little more important than $x_j$ | 3 |
| Strong | $x_i$ is more important than $x_j$ | 5 |
| Very Strong | $x_i$ is very important than $x_j$ | 7 |
| Extreme | $x_i$ is vitally important than $x_j$ | 9 |
| Inner | between above neighboring | 2/4/6/8 |

### 4.2. Update of Node State Table

As mentioned before, the node state value is calculated according to the content transfer history, which is kept in LOG on each edge router. During the continuous content delivery in the network, the NST on each edge router needs to be constantly updated to reflect the latest state information, such as replica node loads and link loads. Otherwise, failure to update information promptly will result in incorrect replica selection decisions, and make the data transmission in network performance worse. The update algorithm of NST based on the information in LOG is presented in Algorithm 2. The update operation is triggered when the edge router obtains the transmission information collected from sending a content request out and receiving a content response. The items in LOG are defined as {*nodeNA, Hops, OSR, RTT, delay$_{queuing}$*}, where *nodeNA* is the address of the replica node; the other items are the same as mentioned in Section 4.1. In the maintenance of LOG, a content transmission flow is labeled by *nodeNA* and *contentID* together, and *RTT* is calculated by the local timestamps at both the send time of a request packet and the receive time of the corresponding response packet.

When the network is built based on an IP-compatible infrastructure, the requested content is split into smaller ICN packets encapsulated with an IP header according to the link maximum transmission unit (MTU) before transmitting to the network, which is used to avoid IP fragmentation operations. In this situation, LOG should not be updated by every ICN packet, because overly frequent update operations may result in high computational overhead and less performance improvement. Therefore, we suggest sampling only the first and last of the content response packet for information collection and NST updating.

Specifically, ERS was designed to operate on top of the ICN solutions with a standalone name resolution approach, while the design principles to select replicas can be more broadly applicable to other ICN frameworks.

---

**Algorithm 2: Update of NST**

---

**Input:** *NST*, *LOG*, *packet*
**Output:** *NST*, *LOG*
**Parameters:** $\vec{w}$
1:   **if** *packet* is a content request **then**
2:         **for** *l* in *LOG* **if** *l.nodeNA* $==$ *packet.destinationNA* **do**
3:             *l.requestTime* $\leftarrow$ *currentSystemTime*
4:             *l.OSR* $\leftarrow$ *l.OSR* $+ 1$
5:             update *NST*[*packet.destinationNA*]*.value* using Equation (6) with $\vec{w}$ and *l*
6:         **end for**
7:   **else if** *packet* is a content response **then**
8:         **for** *l* in *LOG* **if** *l.nodeNA* $==$ *packet.sourceNA* **do**
9:             *l.responseTime* $\leftarrow$ *currentSystemTime*
10:            *l.RTT* $\leftarrow$ *l.requestTime* $-$ *L.responseTime*
11:            *l.RTT*$_{base}$ $\leftarrow$ $\min$(*l.RTT*, *l.RTT*$_{base}$)
12:            *l.OSR* $\leftarrow$ *l.OSR* $- 1$
13:            *l.delay*$_{queuing}$ $\leftarrow$ *l.RTT* $-$ *l.RTT*$_{base}$
14:            update *l.Hops* using the hops field in p*acket*
15:            update *NST*[*packet.sourceNA*]*.value* using Equation (6) with $\vec{w}$ and *l*
16:         **end for**
17:   **end if**
18:   **return** *NST*, *LOG*

---

The information in LOG plays a key role in NST. However, we find that the timeliness of this information may be poor when the request frequency is low. More specifically, an edge router may not receive any feedback information from a replica node for a long time, when there is no request for contents cached on the replica node or the replica node is not selected due to its poor performance. In this condition, the NST value and the LOG information on the edge router about this replica node are not timely, and the previously maintained information is not positive and even has the opposite effect. To this end, we designed a refreshing mechanism, taking into account the dynamic nature of the network. Among the three factors mentioned above, network distance is related to the newest record and has nothing to do with historical information. Thus, only the measurement of path congestion degree may be affected by the historical record. This is to say that the timeliness of queuing delay may become poor when there is no request from an edge router to the replica node for a long time. The queuing time is calculated by the base RTT during past content delivery. By periodically setting the base RTT back to its initial value and calculating it in time, the current network congestion situation can be more accurately reflected. In this paper, we set the base RTT period to be 60 s as introduced in LEDBAT [45].

## 5. Performance Evaluation

This section presents experimental studies on the replica selection approach proposed in this paper. We evaluated the performance of ERS in terms of content retrieval and the load balancing of replica nodes and network links in comparison with several other

approaches. Additionally, the delay range of the ENRS domain and the overhead induced by the ERS method were analyzed.

### 5.1. Experimental Setup

To evaluate our proposal, we implemented ERS based on Icarus [49], which is a lightweight simulator that allows researchers to easily test customized ICN in-caching and replica selection strategies. Detailed performance evaluation and overhead analysis are given below. Table 5 summarizes the main descriptions of the simulation configuration.

**Table 5.** Simulation configuration.

| Parameter | Description |
| --- | --- |
| Topology | TISCALI |
| Catalog | $10^4$ contents with Zipf distribution |
| Traffic workload | Stationary |
| Zipf skewness | *alpha* = 0.9 |
| Size of the content | Uniformed, $C$ = 2 MB |
| Delay range of ENRS domain | 15 ms |
| Cache strategy | LCE |
| Cache replacement | LRU |
| Request number | $2 \times 10^4$ |
| Requests | 1000 req/s with a Poisson distribution |

In the simulation, the stationary workload is tested with a catalog of $N = 10^4$ contents following a Zipf distribution with a default skewness parameter $alpha = 0.9$. The requests of this workload are designed up to $2 \times 10^4$ with the Independent Reference Model (IRM). Each ICN router implements the Least Recently Used (LRU) replacement policy [40]. The cache of each router is empty at the beginning, and the cache size of each node is equal. The first $10^4$ requests are used to allow caches to converge and NSTs to stabilize, and these requests are not used for gathering statistics. The next $10^4$ requests are logged to obtain statistics for analyzing the replica selection method. Each user issues content requests as a Poisson process with a default rate of 1000 req/s. Assume that shortest path routes are computed using Djikstra's shortest-path (lowest-weight) algorithm [42]. The results are obtained by running the experiments 10 times and analyzed with a 95% confidence interval.

In our experiment, content was the same as an object chunk, which is the basic data unit to which naming, transmission, and caching are applied [50–52]. Due to the substantial overhead brought from small-sized chunks [50], we set the size of content to a few MBs, such as 2 MB. The bandwidth was set to 1000 Mbps. To calculate the weights in Equation (6), we set $a_{12} = 5$, $a_{13} = 9$, $a_{23} = 2$ in Table 3. Then, we obtained $a_{21} = 1/5$, $a_{31} = 1/9$, $a_{32} = 1/2$, and $\vec{w} = \{0.757, 0.162, 0.081\}$.

### 5.2. Performance Comparison

An Internet Service Provider (ISP) topology called TISCALI was generated by the Rocketfuel mapping engine [53] to run the simulations. TISCALI depicts the realistic Internet environment of Europe. As shown in Figure 5, this topology consists of 240 nodes and 404 edges, including 27 content sources (red star), 50 cache-supported ICN routers (blue square), 37 receivers (yellow triangle), and 126 IP routers (green circle).

We evaluated the performance of ERS based on three metrics: average content download delay, the standard deviation of replica node load, and the standard deviation of link load. The average content download delay indicates the average delay experienced by users since the departing of a content request and the arrival of the corresponding content. This is among the most important metrics for content retrieval. The load is the number of requests that the replica node or link processed. The standard deviation of replica node load refers to the proportion of content requests responded to by each replica node. This reflects the load balancing between replica nodes. The standard deviation of link load

represents the proportion of requests processed by each link. Here, the standard deviation of the replica node load (link load) is denoted as Equation (8):

$$S = \sqrt{\frac{1}{m} \sum_{i=1}^{m} \left(load_i - \overline{load}\right)^2} \qquad (8)$$

where the total number of replica nodes (links) is $m$, $\overline{load}$ is the average number of requests processed by each replica node (link), and $load_i$ is the number of requests processed by the replica node i (link).

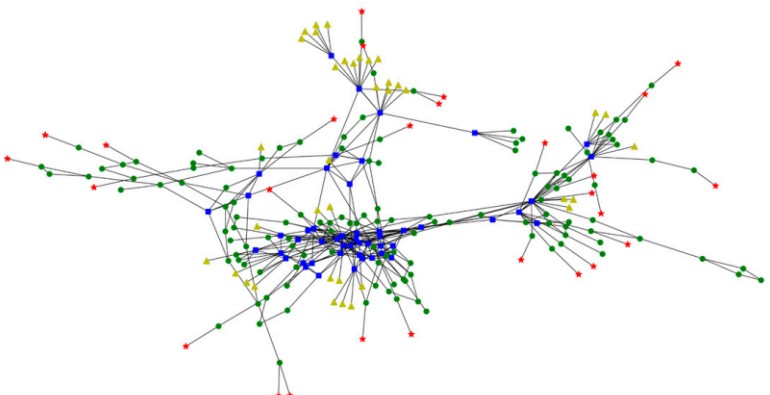

**Figure 5.** TISCALI topology.

Simulation scenarios were altered in terms of three impact factors: cache-to-population ratio ranges from 0.1 to 1.0, request rate ranges from 100 to 1000 req/s, and Zipf parameter alpha ranges from 0.6 to 1.3. The cache-to-population ratio shows the proportion of total cache size to total content size. Zipf parameter alpha indicates the skewness of popularity distribution.

5.2.1. Average Content Download Delay

Content download delay is among the most important indicators of content retrieval performance in the ICN scenario. We set up comparison experiments to analyze the performance of the proposed ERS method in average content download delay. First, we chose three content retrieval architectures different from ERS. MF uses a centralized global name resolution service, and the nearest replica selection strategy (denoted by MF-Nearest). NDN (denoted by NDN-Pure) is an ICN scheme in which content is retrieved and forwarded directly by the hierarchical name and the replica is found by chance on the forwarding path. We also reproduced NDN using the O2CEMF probe mechanism (denoted by NDN-O2CEMF) to represent the modified NDN schemas that could be effectively discovered off-path replicas. Second, we also implemented two different replica selection strategies using the ENRS domain adopted by ERS for comparison: the Random-Intradomain method means to choose one of the replica nodes in the local resolution domain randomly. The main idea of the L2-Intradomain method is to select the replica node with the minimum load first, and then select the nearest node among the nodes with the same load. Even though the L2 mentioned in Section 2.2 is proposed for a distributed key-value database system, we learned its ideas and migrated them to the content retrieval scenario in ICN for comparison of different intradomain replica selection strategies.

Figure 6 shows the average content download delay of each algorithm in the simulation. Figure 6a shows the impact of request rates on content download delay, and we can determine that the ERS method has the smallest content download delay compared to other methods in most cases. NDN-Pure and NDN-O2CEMF show good performance at a low request rate. The NDN-O2CEMF method performs almost uniformly with ERS when the request rate is between 500–700 req/s. This is because the name-based routing under

NDN architecture does not have the delay caused by name resolution, which is a natural advantage over ICN architecture based on the standalone name resolution. However, as a cost, the NDN-Pure schema has poor capability to find off-path replicas. By introducing a probe mechanism, NDN-O2CEMF enhances its ability to find off-path replicas; the content download delay is effectively reduced by spending more communication overhead. With the increase in the request rate, the content download delay tends to increase, because the delay brought by replica node load and link load is increasing. We noticed that the content download delay of NDN-Pure and NDN-O2CEMF grows faster compared with other algorithms. A reasonable explanation is that the methods based on NDN architecture converge more traffic on the path towards the content source, which makes the delay brought by the link load higher. Compared with other algorithms, the Random-Intradomain method has poor performance. The Random-Intradomain replica selection method is not a reasonable choice, even if Random-Intradomain does not have any additional information maintenance overhead. As we can see from Figure 6b, with the increase in cache size on replica nodes in the network, more requests can be hit in the nearest replica nodes, and the content download delay of each algorithm presents a downward trend. A larger Zipf parameter alpha means that content requests in the network are more concentrated, so it leads to the higher cache hit ratio of replica nodes, and the correspondingly lower content download delay, which is consistent with the result of the simulation shown in Figure 6c. In addition, we also found that the degree of concentration of content requests has a greater impact on the NDN-based approach. The reason for this is that the benefits of caching efficiency improvement along the path from the node to the content source are more obvious in NDN.

In conclusion, ERS has a smaller content download delay among all algorithms in our simulation scenarios, which fully demonstrates that ERS has excellent content retrieval performance by using a local limited domain to discover replica nodes and integrating three factors to select replicas.

### 5.2.2. Standard Deviation of Replica Node Load

Balancing the service load of replica nodes can effectively avoid performance degradation caused by replica node overload. The standard deviation of replica node load is used to reflect the equilibrium of the replica nodes providing the content service. The replica node load represents the number of requests served by this node. We set up an experiment to study the load balancing of the methods based on the standalone name resolution, including ERS, MF-Nearest, L2-Intradomain, and Random-Intradomain. Similarly, we also explored the effects of the request rate, cache-to-population ratio, and Zipf parameter alpha. The simulation results are shown in Figure 7.

Figure 7 indicates that the performance at the replica node load aspect of ERS is only worse than the Random-Intradomain method. This is because the Random-Intradomain method does not have a priority principle when selecting replica nodes. The MF-Nearest approach is worst at service balancing, because it always selects the nearest replica node, resulting in a much higher probability that the replica node near the receiver in the network topology is selected than the other copies. ERS adopts a more granular approach to the selection of replicas than the L2-Intradomain, which can receive the negative feedback of high-load replica nodes to avoid choosing these as content senders, resulting in providing a better balance between replica nodes.

From Figure 7a, we can see that as the request rate increases, the equilibrium of ERS and L2-Intradomain changes more smoothly than that of MF-Nearest and Random-Intradomain. This indicates that the state maintenance of replica node load in ERS can effectively prevent overloaded nodes from being selected. In Figure 7b, with the increase in the cache size, the load of the replica node declines at first and then starts to rise. The load first drops because the cache hit ratio is low when the cache size is small. At this time, most of the content will be served by content sources, and it causes loads of content sources to be very high. Then, the hit ratio of replica nodes in the network increases rapidly with the increase in cache size. The cache-supported routers can alleviate the load of the content

source, and the equilibrium will be improved. However, each algorithm has a tendency to choose the closer copy. When the cache size becomes higher, the equilibrium deteriorates because the replica node near the receiver will be selected more frequently. The degree of request concentration corresponding to the alpha in Figure 7c is also closely related to the cache hit ratio. It also proves that the equilibrium deteriorates with the increase in cache size.

In sum, without considering the Random-Intradomain method, the ERS strategy can effectively improve the efficiency of overall content retrieval by balancing the load of replica nodes.

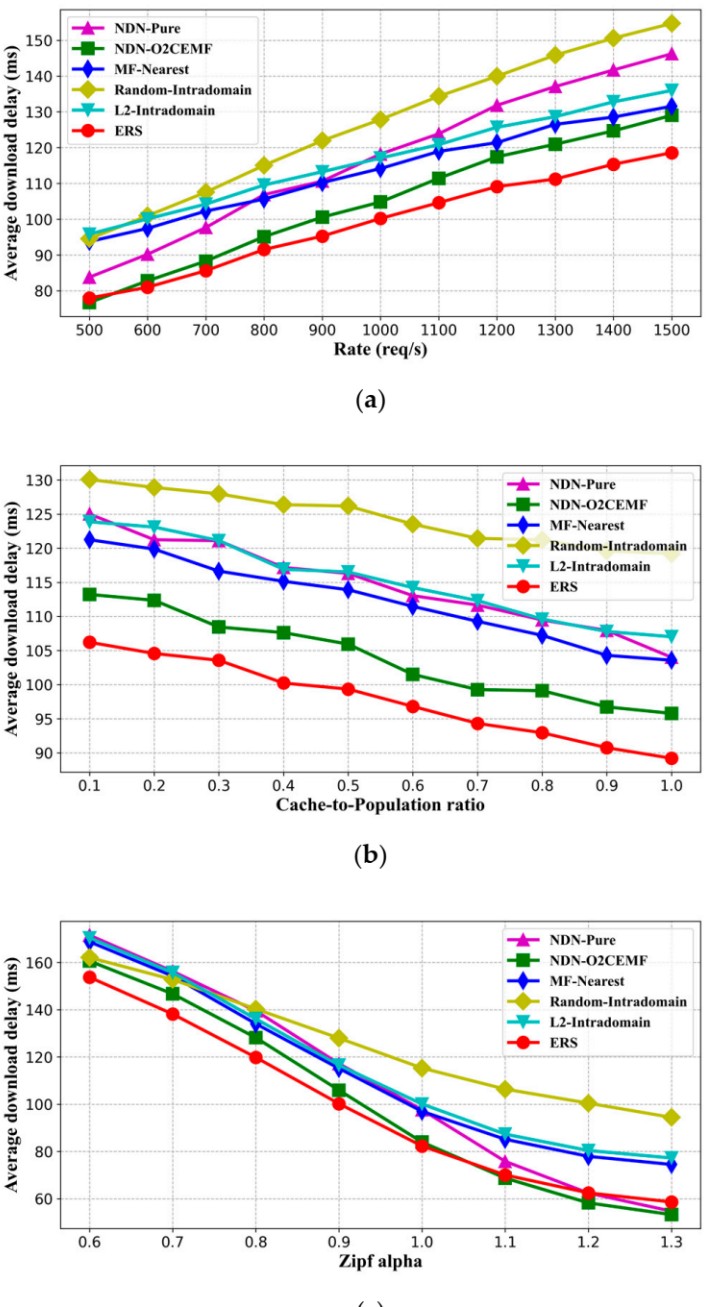

**Figure 6.** Average content download delay comparison with different impact factors. (**a**) Request rate, (**b**) cache-to-population ratio, and (**c**) Zipf parameter alpha.

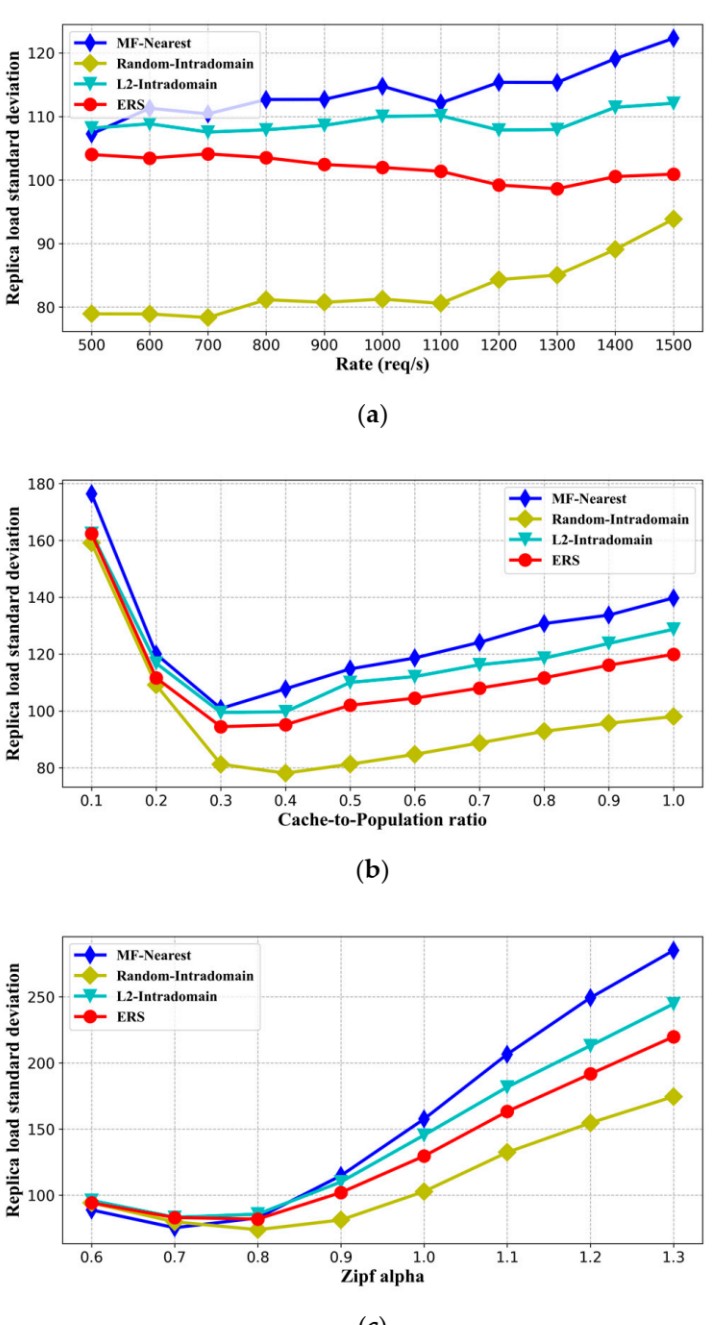

**Figure 7.** The standard deviation of replica node load comparison with different impact factors. (**a**) Request rate, (**b**) cache-to-population ratio, and (**c**) Zipf parameter alpha.

### 5.2.3. Standard Deviation of Link Load

In this paper, we introduced the concept of path congestion degree in the ICN replica selection problem for the first time. The path congestion degree was used to avoid the problem of router queuing for too long caused by network congestion in some links, such as bottleneck links. We used the standard deviation of link load as a measurement index to observe whether the proposed ERS method avoids the congested paths in the process of content retrieval. The results are shown in Figure 8; the load balancing between links under the ERS method is better than that of other algorithms. This indicates that the ERS can avoid the congested path to a certain extent by taking the congestion degree as a calculation factor for replica selection. Additionally, we noted that the Random-Intradomain strategy performs better in terms of replica node equilibrium in Figure 7. However, the random

selection method makes content retrieval more likely to pass through bottleneck links, resulting in a bad balance of link load. Combining the above analysis and comparing ERS with other algorithms, we can conclude that the path congestion degree considered in ERS is beneficial to the performance of content retrieval in ICN.

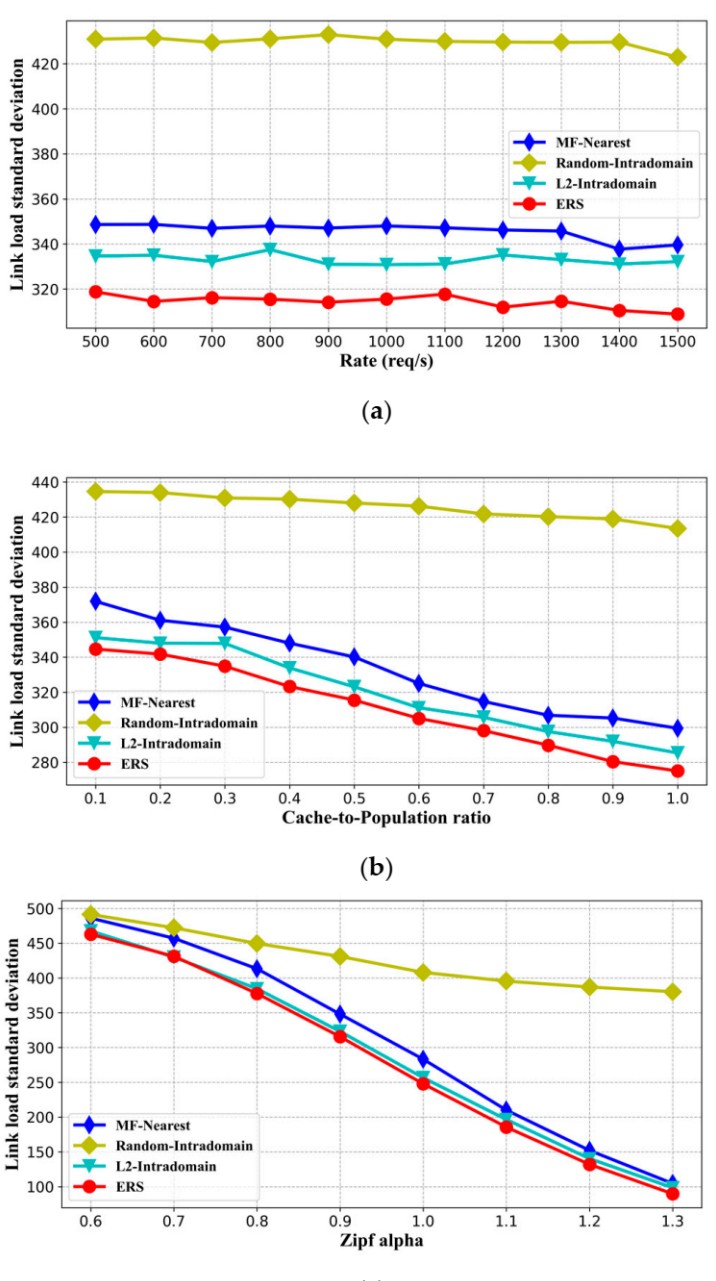

**Figure 8.** The standard deviation of link load comparison with different impact factors. (**a**) Request rate, (**b**) cache-to-population ratio, and (**c**) Zipf parameter alpha.

In addition, the influence trend of each impact factor on the balance of link load can also be revealed according to the experimental results. Figure 8a shows that, for the algorithms conducted in the experiment, the balance of data transmission flow in each link is basically stable at different request rates. As shown in Figure 8b,c, with the increase in network cache or the concentration of content requests, transmission traffic becomes more balanced on network links. This is because the larger cache-to-population ratio or more focused content requests can effectively improve the edge replica hit rate near the user, to

avoid a large amount of traffic through the bottleneck link of the network, so as to improve the balance.

### 5.2.4. Real-World Dataset

We also utilized the dataset of YouTube Video traffic to further verify the performance of our proposed method compared with others. The YouTube Video traffic records a total of $3.8 \times 10^6$ requests [54] for $1.76 \times 10^6$ videos, coming from 31,124 distinct IP addresses. Half of the total requests were used to warm up the network, and the remaining were used to gather statistics. Since the distribution of content requests has been fixed in the real-world data set, we only conducted comparative experiments on request rate and cache-to-population ratio. Figure 9 indicates that the performance of each algorithm running with the YouTube Video traffic dataset is basically the same as that of the stationary traffic shown in Figure 6. Different from the Zipf request distribution shown in Figure 6, we found that the NDN-Pure method declined more slowly as the cache-to-population ratio increased compared to other methods. This indicates that the NDN-Pure method is not making sufficient use of the cache in the YouTube dataset. However, the ERS method can still achieve the shortest average download time for the real traffic represented by this data set. Therefore, this experiment further proves the effectiveness of the ERS method for replica selection under different flow characteristics.

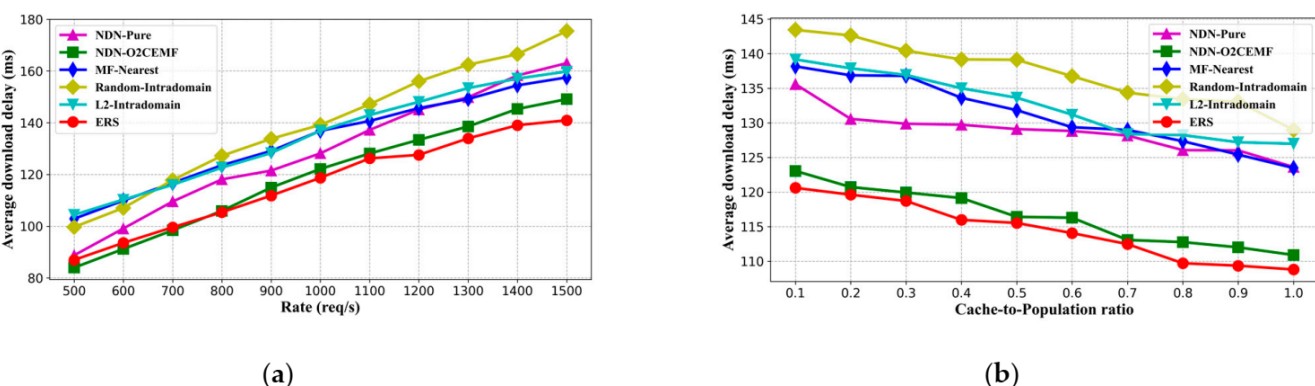

**Figure 9.** Average content download delay comparison with different impact factors on YouTube video traffic dataset. (**a**) Request rate and (**b**) cache-to-population ratio.

### 5.3. Delay Range of ENRS Domain

We investigated the impact of the delay range of the ENRS domain on the performance and overhead of ERS. The delay range of the ENRS domain was set from 5 to 25 ms with a step size of 1ms. In Figure 10, the intradomain hit ratio is defined as the ratio of the number of requests processed by intradomain replica nodes to the total number of requests. The NST-to-ReplicaCount ratio (the ratio of the entries of NSTs to the replica nodes count) can indicate the maintenance overhead of NST to some extent.

Figure 10 shows that the intradomain hit ratio and the NST-to-ReplicaCount ratio first increased as the ENRS scope increased and then tended to be stable when the delay range reached 22.5 ms. This is because the number of domains divided by ENRS gradually dwindled with the increase in the delay range, and there was only one ENRS domain in the overall network when the delay range reached 22.5 ms. Accordingly, as the scope increased, the content download delay of ERS decreased slowly. Then, from the trade-off of performance improvement and information maintenance overhead, we set the delay range of the ENRS domain to 15 ms in the above simulations of ERS. When the delay range of the ENRS domain was 15 ms, the intradomain hit rate was about 60%, and the information maintenance overhead was about 30% of the overall network. The delay range of the ENRS domain can be determined more accurately by using an intelligent approach, such as machine learning.

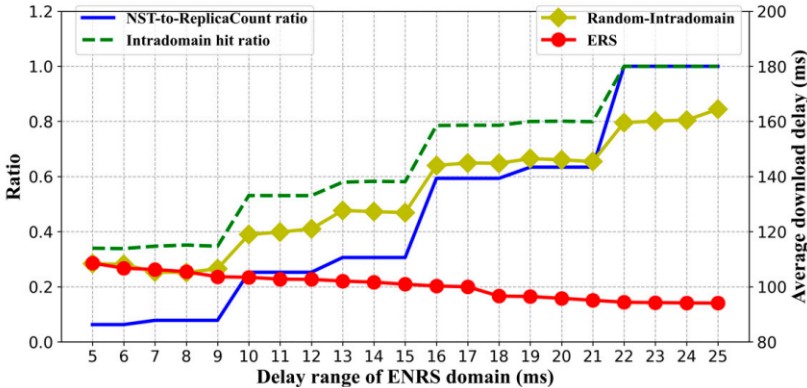

**Figure 10.** Performance and overhead under different delay ranges of ENRS domain.

As mentioned in Section 3.3, the random method is recommended to adopt when the delay range of ENRS domain is small. This is because the random method is easy to implement and has low overhead. Therefore, the variation of the content download delay of the Random-Intradomain method with the increase in the delay range of the ENRS domain is also plotted in Figure 10. We can see that the average content download delay of ERS is almost the same as that of Random-Intradomain when the delay range of the ENRS domain is less than 9 ms, because there are only 1–3 replica nodes in each domain. Therefore, as mentioned in Section 3.3, a simple random replica selection strategy can be used when the delay range of the domain is small, i.e., less than 9 ms.

### 5.4. Overhead Analysis

The ERS method proposed in this paper improved the performance of the network, but it also brings additional overhead. As we can see from Section 3, the ERS approach introduces two mechanisms; one is a mixture name resolution scheme combining ENRS and GNRS, and the other is the usage of NST. Next, we will analyze the overhead from these two aspects.

Authors in [55] confirmed that the cost of updating the control layer is much lower than the cost of data download, so resolution-based content discovery can be a good solution. As shown in Figure 11, the average resolution hops for each content retrieval of ERS is about decreased by 9.96% compared to MF. Furthermore, authors in [19] claimed that the GNRS update traffic only takes up a minute fraction of the overall Internet traffic. Thus, the name resolution traffic introduced by the ERS is acceptable. As described in Section 4, the state of replica nodes is obtained by recording historical transfer information, rather than active probing, which avoids introducing additional traffic overhead. From the experiment results in Figure 10, ERS can achieve a high in-domain hit ratio at a lower maintenance cost, so maintaining the data structure of the NST table in a reasonable range of domains does not cause a significant burden on memory.

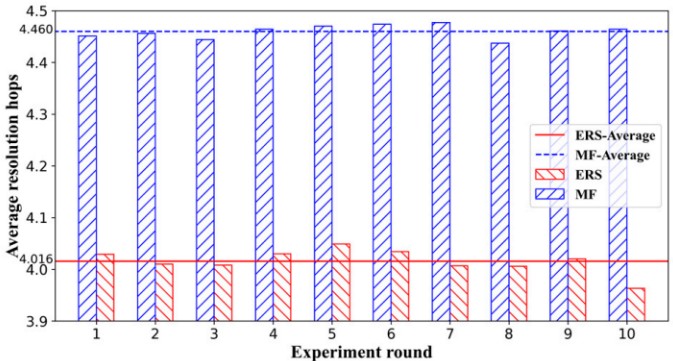

**Figure 11.** Average resolution hops for each content retrieval.

In summary, the ERS method not only performs better in terms of content download delay than other replica selection methods but also has no significant overhead increase. Therefore, our proposed ERS method is effective.

## 6. Conclusions

In this paper, we discussed the efficient content retrieval problem in ICN scenarios. To solve this problem, an enhanced replica selection method called ERS is proposed. This method first uses a distance-constrained-based name resolution system to discover the nearby replicas, and then, the most appropriate replica is chosen according to a local state table that maintains the state of replica nodes within a limited domain. We considered three main factors to evaluate the node state for replica selection, network distances, replica node loads, and the innovatively introduced path congestion degree. The detailed method to collect the information of these factors and update the node state table was also designed. We compared our method with several state-of-the-art methods on stationary traffic and a real-world dataset to fully measure the performance of our proposed method. The results demonstrated that our proposed ERS approach outperforms other existing approaches in terms of delivering content to users with a small download delay. It also performs effectively in the load balancing of replica nodes and links, which proves the effectiveness of the proposed method. Finally, the delay range of the ENRS domain and the overhead of the ERS method are also discussed, and the feasibility of our method is illustrated.

In the future, our research direction will focus on the further optimization of the method for replica selection in ICN. We will try to introduce intelligent methods, such as machine learning algorithms, to make full use of the information collected in data transmission, so as to realize more accurate replica selection.

**Author Contributions:** Conceptualization, Y.S., H.N. and X.Z.; methodology, Y.S., H.N. and X.Z.; software, Y.S.; writing—original draft preparation, Y.S.; writing—review and editing, Y.S., H.N. and X.Z.; supervision, X.Z.; project administration, X.Z.; funding acquisition, H.N. All authors have read and agreed to the published version of the manuscript.

**Funding:** This work was funded by the Strategic Leadership Project of Chinese Academy of Sciences: SEANET Technology Standardization Research System Development (Project No. XDC02070100).

**Institutional Review Board Statement:** Not applicable.

**Informed Consent Statement:** Not applicable.

**Acknowledgments:** We would like to express our gratitude to Jinlin Wang, Jiaqi Li, and Rui Han for their meaningful support in this work.

**Conflicts of Interest:** The authors declare no conflict of interest.

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
