# Peer review of "An Enhanced Replica Selection Approach Based on Distance Constraint in ICN"

_electronics, doi:10.3390/electronics10040490_

Round 1
Reviewer 1 Report
- The figures in the paper are small and do not have a good resolution. I recommend changing the figures.
- The results obtained and presented in the figures in the article require more comments.
- I recommend the presentation of the values from table 5 in graphical form, with the highlighting of the average values.
- I recommend extending the final conclusions (they must be much more punctual).
Reviewer 2 Report
This paper proposes an Enhanced Replica Selection Approach Based on Distance Constraint in ICN. It is a replica selection approach called ERS in ICN. ERS improved content retrieval efficiency by using a content discovery mechanism based on distance constraint and a replica selection strategy based on a node state table. The Experimental results show that the proposed approach has a good performance.
However, the writing of the paper needs some improvements for publication.
The related work is too short and week, I recommend the author add a summary table after the related work section that compares all of the cited approaches based on the metrics discussed on the evaluation section.
The dealy range of the ENRS domain and the overhead of the ERS method
can be improved if the authors use an Intelligent approach, such as machine learning and reinforcement learning techniques.
In the evaluation section, the author should justify why the chosen baseline for comparison. The proposed method should be compared with at least another method from the literature.
Overall, The proposed solution can open the theory to practical implementation that can give us the solution in a realistic amount of time. The suggestions given in the review are not mandatory, but can be accommodated in a possible revised version in case one is made.
